# Numerical Prediction of Microstructure Evolution of Small-Diameter Stainless Steel Balls during Cold Skew Rolling

**DOI:** 10.3390/ma16083246

**Published:** 2023-04-20

**Authors:** Jing Zhou, Shengqiang Liu, Baoyu Wang, Hao Xu

**Affiliations:** 1School of Mechanical Engineering, University of Science and Technology Beijing, Beijing 100083, China; 2Engineering Research Center of Part Rolling, Ministry of Education, Beijing 100083, China

**Keywords:** steel ball, skew rolling, microstructure, cold forming, stainless steel

## Abstract

The wear resistance and hardness of stainless steel (SS) balls formed by cold skew rolling are effectively improved due to the change in internal microstructure. In this study, based on the deformation mechanism of 316L stainless steel, a physical mechanism-based constitutive model was established and implemented in a subroutine of Simufact to investigate the microstructure evolution of 316L SS balls during the cold skew rolling process. The evolution of equivalent strain, stress, dislocation density, grain size, and martensite content was studied via simulation during the steel balls’ cold skew rolling process. The corresponding skew rolling experiments of steel balls were carried out to verify the accuracy of the finite element (FE) model results. The results showed that the macro dimensional deviation of steel balls fluctuates less, and the microstructure evolution agrees well with the simulation results, which proves that the established FE model has high credibility. It shows that the FE model, coupled with multiple deformation mechanisms, provides a good prediction of the macro dimensions and internal microstructure evolution of small-diameter steel balls during cold skew rolling.

## 1. Introduction

Steel balls, as one of the critical components of ball bearings, directly influence the dynamic performance, reliability, and life of the bearings. Numerous researchers have conducted in-depth research on the steel ball skew rolling process, which has dramatically deepened the understanding between the microstructure and performance of steel balls. However, most of them are limited to the hot skew rolling process in the formation of steel balls with larger diameters, and there needs to be more research on steel balls with small diameters. The forming of small-diameter steel balls is more complicated than that of large-diameter steel balls, the equipment and process parameters required are stricter and more precise, and the forming defects and dimensions of the steel balls are more challenging to control rigorously. In addition, most of the research on steel ball materials is concentrated on the steel GCr15, while more research needs to be conducted on stainless steel (SS) balls. Compared with GCr15, 316L SS exhibits better rust and corrosion resistance [1]. Bearings manufactured with SS can be used in liquids and have stronger heat resistance and are widely used in medical devices, cryogenic engineering, optical engineering, high-speed motors, and other fields.

Several studies have examined the influence of process parameters on the steel balls’ skew rolling forming process. Pater et al. [2] discussed the production process of the multi-wedge spiral rolling process in the formation of steel balls. They analyzed the temperature, stress, strain, force, and movement during the forming process. Gontarz et al. [3] studied the effect of skew groove patterns on obliquely rolled steel balls and analyzed the strain, damage, and temperature distribution during the forming process. Tomczak et al. [4] optimized the spiral indentation of the die during the oblique rolling of steel balls, proposed a design method for the spiral indentation, proposed a die correction method based on it, and verified the effectiveness of the optimized die method by simulation and forming experiments. Liu et al. [5] analyzed the production process of small-diameter steel balls, discussed the influence of process parameters on the forming quality of steel balls, and analyzed the stress, strain, rolling force, and work-hardening distribution of steel balls, and successfully trial-produced small-diameter steel balls with the required dimensional accuracy. The above studies mainly focus on the hot skew rolling process of steel balls, while steel balls with small diameters are more suitable for cold forming due to the small size and fast heat loss during the hot rolling process. Moreover, the macroscopic dimensional fluctuations and internal microstructure evolution during the cold skew rolling of steel balls have an important influence on their performance and service life, so research on the cold skew rolling process of steel balls is urgently needed.

During deformation, the deformation mechanism of austenitic SS is highly dependent on the stacking faults energy (SFE), which is the crucial parameter determining whether mechanical twinning, martensitic transformation, and dislocation slipping dominate the deformation process of the material. The deformation mechanism of metals with low SFE will change from dislocation slip to deformation twinning or martensitic transformation, which is significant for the hardening of the material. For the deformation mechanism of austenitic SS under rolling conditions, Zhang et al. [6] investigated the strain-induced martensite behavior of 316L SS during rolling at different temperatures. After warm rolling deformation, they found that both slate-like martensite and dislocation-type martensite were present in the microstructure, containing both dislocations and deformation twins. Xiong et al. [7] similarly studied the evolution of microstructure and mechanical properties of 316LN SS after cold rolling at different strains. They found that the deformation mechanisms during low-temperature rolling were mainly composed of high-density dislocations, deformation twinning, and martensitic transformation. Eskandari et al. [8] investigated the changes in the deformation-induced martensite volume fraction and mechanical properties of 316L SS under cold rolling process conditions, which showed that decreasing the rolling temperature and increasing the pre-strain could increase the martensite volume fraction and, at the same time, refine the grain size, with a minimum of 30–40 nm grain size. The obtained nanocrystalline SS exhibited excellent tensile strength.

The constitutive model based on the physical mechanism enables reflection of the connection between the macroscopic deformation behavior and the material’s microstructure. Huo et al. [9] studied the forming process of bearing steel balls with a diameter of 30 mm during warm skew rolling. They developed an internal state variable constitutive model to predict the microstructure evolution. Then, the experimental and simulation results verified the availability of the developed model. Ran et al. [10] established a multiscale model with coupled damage, grain size, and surface layer models to describe the ductile fracture and deformation behavior during the micro-forming of multiphase alloys and elucidated the relationships between size effects, fracture energy, and expected fracture strain during the micro-forming. Chen et al. [11] developed a constitutive model considering the kinetics of twinning and martensitic transformation using the mixing law to investigate the impact processes during the mechanical abrasion treatment of surfaces. Liu et al. [12] established a constitutive model coupling the electricity and size effect, studied the influences of current density and grain size on the forming process in the electro-assisted forming, and captured the abnormal evolution of surface effect by using the proposed constitutive model.

This work aims to implement a physical mechanism-based constitutive model coupling dislocation slip, martensitic transformation, and grain refinement in the finite element software Simufact 16.0. Establishing the FE model of the small-diameter steel ball skew rolling process is conducive to investigating the macro size and microstructure evolution of cold-rolled steel balls and provides a theoretical basis for the microstructure control in the process of cold skew rolling of steel balls.

## 2. Development of a Mechanism-Based Constitutive Model of 316L under Cold Deformation

### 2.1. Establishment of Multiscale Constitutive Equations

As the martensite transformation of 316L stainless steel will occur during cold plastic deformation, the microstructure after deformation consists of austenite and martensite. Therefore, the mixture law describes the macroscopic flow stress of the dual-phase material. The macroscopic flow stress can be expressed in the following formula:(1)$$\sigma =\sigma_{A}(1-f_{M})+\sigma_{M}f_{M}$$
where $\sigma_{A}$, $\sigma_{M}$ are the stress of austenite and martensite, and the stress of martensite can be calculated by:(2)$$\sigma_{M}=\sigma_{0}+\alpha MG_{M}b_{M}\sqrt{\rho_{M}}$$
where $\sigma_{0}$ is the yield strength of martensite. α, M, $G_{M}$, $b_{M}$, and $\rho_{M}$ are the material constants reflecting the dislocation interaction, the Taylor factor, the shear modulus, the magnitude of the burger vector, and the dislocation density of martensite, respectively. The value can be determined by the empirical formula [13]:(3)$$\sigma_{0}(MPa)=461+1310\times \sqrt{\left(wt.\%C\right)}$$

Austenitic stress is composed of grain boundary resistance $\sigma_{G}$ and dislocation density resistance $\sigma_{p}$ [14]:(4)$$\sigma_{A}=\sigma_{G}+\sigma_{\rho }$$

$\sigma_{G}$ can be expressed by the Hall–Petch equation, and $\sigma_{\rho }$ is related to mean dislocation density [15]:(5)$$\sigma_{G}=\sigma_{y_{0}}+\frac{K_{s}}{\sqrt{d}}$$
(6)$$\sigma_{\rho }=\alpha MG_{A}b_{A}\sqrt{\rho_{A}}$$
where $\sigma_{y_{0}}$ is the initial stress, $K_{s}$ is the parameter indicating the Hall–Petch slope, and *d* is the grain size. The evolution of dislocation density in austenite can be denoted as:(7)$${\dot{\rho }}_{A}=\frac{\left|\dot{\varepsilon }\right|}{b_{A}\Lambda_{s}}-\Omega \rho_{A}\left|\dot{\varepsilon }\right|$$
where $\dot{\varepsilon }$ is the plastic strain rate, and Ω is the content describing dynamic recovery. $\Lambda_{s}$ is the mean free path of dislocation density in austenite, which is related to grain size, dislocation density, and martensite volume fraction:(8)$$\frac{1}{\Lambda_{s}}=\frac{1}{k_{d}d}+\frac{\sqrt{\rho_{A}}}{k_{s}}+\frac{f_{M}}{k_{M}t_{M}(1-f_{M})}$$
where $k_{d}$, $k_{s}$, $k_{M}$ are material constants, and $t_{M}$ is the thickness of the martensite lath; $f_{M}$ is the volume fraction of martensite.

In this paper, the martensitic transformation model proposed by Wong et al. [16] is adopted. It is believed that the volume fraction of martensite is related to the martensite nucleation rate ${\dot{N}}_{M}$ and the volume of newly formed martensite $V_{M}$:(9)$${\dot{f}}_{M}=(1-f_{M})V_{M}{\dot{N}}_{M}$$

${\dot{N}}_{M}$ can be expressed by the following equation:(10)$${\dot{N}}_{M}={\dot{N}}_{0}P_{M}P_{nsc}$$
where ${\dot{N}}_{0}$ is the potential martensitic nucleation density in unit time, $P_{nsc}$ is the probability that cross-slip does not occur. $P_{M}$ is the probability of the formation of a martensite lath:(11)$$P_{M}=\exp[-{\left(\frac{\sigma_{tr}^{c}}{\sigma_{M}}\right)}^{m}]$$
where *m* is the model parameter, $\sigma_{tr}^{c}$ is the critical stress of martensitic transformation, and $\sigma_{M}$ is the stress applied to the martensite.
(12)$$\sigma_{tr}^{c}=\sigma_{tr}^{0}+\frac{K_{tr}}{\sqrt{d}}$$
where $\sigma_{tr}^{0}$, $K_{tr}$ are the model parameters. The volume of the newly formed martensitic slats can be characterized as:(13)$$V_{M}=\frac{\pi }{4}\Lambda_{M}^{2}t_{M}$$
where $\Lambda_{M}$ is the mean free path of the dislocation in the martensite phase.
(14)$$\frac{1}{\Lambda_{M}}=\frac{c_{\gamma }}{d}+c_{m}\sqrt{\rho_{M}}$$
where $c_{\gamma }$ and $c_{m}$ are the model parameters. The change rate of dislocation density in martensite is given [17]:(15)$${\dot{\rho }}_{M}=\left|\dot{\varepsilon }\right|(\frac{1}{b_{M}\Lambda_{M}}-k_{a}\rho_{M})$$
where $k_{a}$ is the parameter representing the dislocation annihilation. During plastic deformation, the grain of material will be refined, and the grain refinement model can be expressed as: (16)$$d_{\varepsilon }=d_{0}\frac{\left[{\left(\frac{K}{\sqrt{\rho_{A}}}\right)}^{d_{a}}+{\left(1-f_{M}\right)}^{d_{b}}\right]}{2}$$
where $d_{a}$ and $d_{b}$ are the model parameters, $d_{0}$ is the initial grain size, and K is the model parameter.

### 2.2. Solution of Model Parameters

The model is composed of highly coupled differential equations, and there are a total of 24 parameters, so analytical solutions can hardly be obtained. Therefore, the genetic algorithm toolbox in the software package Matlab 2021b is used to solve the model parameters with the goal of residual error between experimental and predicted values. The details of the solution procedure can be found in Ref. [18], and the determined model parameters are given in Table 1.

## 3. Validation of Numerical Simulation of Cold Skew Rolling

### 3.1. FE Modeling

A skew rolling process FE model considering the physical mechanism-based constitutive model of steel balls is developed in the FE software Simufact 16.0, as shown in Figure 1. The FEM consists of two rolls, two guide plates, a guide pipe, and a workpiece. Due to the little plastic and elastic deformation, the guide plates and rolls are set as rigid bodies. The workpiece used in the simulation is the 316L SS wire with a diameter of 3 mm. The density, Young’s modulus, and Poisson’s ratio of 316L SS are 7966 kg/m^3^, 192 GPa, and 0.3, respectively, and its chemical composition is listed in Table 2. In this work, the workpiece is assumed to be a homogeneous isotropic plastic body, the mesh type is hexahedral, and the number of elements is approximately 19,300. The Coulomb friction model is used, the friction coefficient between the rolls and guide plates is 0.2, while it is 0.1 for the friction coefficient between the rolls and the workpiece [19]. In addition, the initial grain size of 316L SS is 16μm, and the initial dislocation density of martensite and austenite are 1 × 10^10^ m^−2^ and 1 × 10^12^ m^−2^, respectively.

### 3.2. Experiment Detail

An experiment of steel ball skew rolling was carried out on the newly designed skew rolling mill, as shown in Figure 2. Two servo motor systems drive the rolls of the skew rolling mill. Before the rolling experiment, 2 rolls deflect the same angle around the axis but along the reverse directions, and the deflection angles are nearly 2 and −2 degrees, respectively. During the rolling process, two rolls rotate in the same direction and drive the workpiece forward, and the workpiece gradually forms steel balls under the extrusion of roll grooves.

### 3.3. Comparison of Steel Ball Diameter

A total of 20 steel balls were randomly selected from the finished steel balls and their arbitrary section diameters were measured by the SZM-45T1-560H stereomicroscope; the results are shown in Figure 3. It can be seen from the figure that the diameter of steel balls ranges from 3.114 to 3.126 mm. The mean square error of the selected balls’ diameter is calculated to further evaluate the dimensional accuracy of the steel ball.

The standard size of the steel ball diameter is 3.12 mm, and then the mean square deviation can be obtained. The calculated mean square deviation from the selected steel balls is 0.004, which means the diameter has less data dispersion, indicating that the macroscopic size of the steel balls has high accuracy. As shown in Figure 3b, the diameter of the steel balls obtained from the simulation is 3.159 mm, close to the standard size. This indicates that the established FE model has good prediction accuracy for the macroscopic size of cold-skew-rolled steel balls.

### 3.4. Comparison of Steel Ball Microstructure

In order to verify the predictive accuracy of the developed FE model based on the dislocation density of the microstructure of the steel ball, four points on the transverse section of the steel balls are selected for microstructure characterization, and the exact positions are also taken in the simulation results. Initially, the steel balls were cut along the transverse section, and one half was selected for mechanical polishing, followed by electrochemical polishing in a 25% perchloric acid alcohol solution for 30 s. Finally, EBSD observation was performed on different positions of the steel ball, and the results were analyzed by HKL Channel 5 software. The comparison between the experimental and simulation results is shown in Figure 4. As can be seen from the figure, the grain size gradually decreases from the center to the surface, and the simulation results are in good agreement with the experiment. Due to the large deformation on the surface of the steel ball, the grain refinement phenomenon is more prominent, while the grain in the center is relatively coarse due to the small deformation.

Figure 5 shows the comparison of the volume fraction of martensite between the simulation and experiment. Because of the randomness of EBSD measurement results, XRD was used to analyze the martensite content of steel balls quantitatively. The martensite content of the transverse sections of three steel balls was measured to ensure the accuracy of the results. The martensite content was determined by X-ray (Rigaku Ultima IV) equipped with a ${\mathrm{Cu}-K}_{\alpha }$ lamp with a scanning angle range of 30~110° and a step setting of 0.02°. In addition, ${\left(111\right)}_{\gamma }$, ${\left(200\right)}_{\gamma }$, and ${\left(220\right)}_{\gamma }$ diffraction peaks of austenite and ${\left(110\right)}_{\alpha^{'}}$, ${\left(200\right)}_{\alpha^{'}}$, and ${\left(211\right)}_{\alpha^{'}}$ diffraction peaks of martensite were selected to calculate the content of martensite by the following equation [20]:(17)$$V_{\alpha }=\frac{1/n\sum_{j=1}^{n}I_{\alpha^{'}}^{j}/R_{\alpha^{'}}^{j}}{1/n\sum_{j=1}^{n}I_{\alpha^{'}}^{j}/R_{\alpha^{'}}^{j}+1/n\sum_{j=1}^{n}I_{\gamma }^{j}/R_{\gamma }^{j}}$$
where *n*, *I*, and *R* are the number of diffraction peaks, intensity factor, and material scattering factor of the corresponding phase, respectively. The results show that the simulation results agree with the experimental results, indicating that the developed FE model has a good prediction accuracy on the martensitic transformation during the cold skew rolling process of steel balls.

Figure 6 shows the comparison of dislocation density at different positions in the steel ball transverse section between experimental and simulation results. The simulation and experimental measured dislocation densities are of the same magnitude, and the values are close, so the FE model can be considered to have good prediction accuracy. It should be noted that the dislocation density obtained from the simulation is the average dislocation density, which is calculated by mixing the austenite and martensite dislocation densities and the volume fraction of each phase according to the mixing rule. Since the developed model is based on the dislocation density, the comparison results of the dislocation density further demonstrate the model’s reliability in predicting the microstructure evolution of steel balls during the skew rolling process.

## 4. Numerical Simulation Analysis of the Skew Rolling Process

### 4.1. Equivalent Strain and Stress of Small-Diameter Steel Balls

The equivalent strain and stress in the longitudinal section of the steel ball during skew rolling are illustrated in Figure 7. During the ball-forming process, the radial compression occurs at the linking neck under the action of the roll ridges. The diameter of the linking neck gradually decreases, and the diameter of the steel ball in the groove gradually increases. The equivalent strain at the linking neck is the largest, while the equator part of the steel ball has a minor strain due to the smaller deformation than other parts. When the workpiece is knifed into the roll grooves, the equivalent stress on the part of the steel ball in contact with the convex edge is the most extensive. Due to the different amounts of deformation, the equivalent stress at the pole part is greater than the equator part in the sphere. The equivalent stress decreases gradually from the outer layer to the inner layer, and the overall stress value of the steel ball is moderate.

Figure 8 illustrates the equivalent strain and stress in the transverse section of the steel ball. The outer surface of the steel ball has the most significant strain. In contrast, the ball’s core has a minor strain because the ball’s surface layer receives the most considerable deformation, and the deformation amount decays from the ball’s surface to the core. The deformation penetration effect on the ball core is insignificant, finally showing the gradual decrease in equivalent strain from the ball surface to the center. It can be found from Figure 8b that the stress distribution in the center is also less than that in the surface, and the area with maximum stress is located at the contact area with the rolls.

### 4.2. Microstructure of Small-Diameter Stainless Steel Balls

Figure 9 shows the dislocation density of martensite and austenite of the steel balls during the skew rolling process. Since the dislocation density of austenite is related to the deformation, the distribution pattern of the dislocation density of austenite is identical to that of strain. The linking neck between the balls has the most significant dislocation density due to the most severe deformation. The part close to the linking neck also has an amount of dislocation density. On the transverse section, the austenitic dislocation density gradually increases from the ball center to the surface. The martensitic dislocation density is not solely related to deformation, which is also related to grain size, and the refined grains will have an inhibitory influence on the martensitic transformation [21]. Therefore, the difference in martensite dislocation density from the center and surface of the ball is relatively slight.

A total of five testing points were selected on the transverse and longitudinal sections of the steel ball to measure the microhardness at different positions of the ball. The testing results are shown in Figure 10. It can be seen that the microhardness continuously increases from the ball center to the surface in both the transverse and longitudinal sections. The only difference is that the variation in the longitudinal section exceeds that in the transverse section. The hardness is highest at point C near the linking neck, and the results are consistent with the dislocation density distribution pattern. The high hardness of the steel ball surface can effectively improve the wear resistance of the steel ball, while the softer part of the ball’s center can improve the impact resistance.

The grain size of the steel balls during the skew rolling process is shown in Figure 11. Because grain refinement is directly correlated to the amount of deformation, the grain size is smaller in the area with more significant deformation. The refined grains on the surface of the steel balls effectively improve the steel balls’ comprehensive mechanical properties and corrosion resistance.

Figure 12 depicts the martensite volume fraction of the steel balls during the skew rolling process. In the transverse section of the steel ball, because the surface grain refinement is more noticeable than the core grain refinement, it will inhibit the formation of martensite. Hence, the martensite content of the surface layer is low, and the center part is slightly high, but the distinction is very slight. In the longitudinal section of the steel ball, due to the deformation of the linking neck part of the steel ball being the largest, the grain refinement is also the most significant, resulting in a reduction in grain size, which will lead to the inhibition of martensitic phase transformation [22]. Meanwhile, because the deformation of the center part is small, the grain size is relatively large. Then, martensite is more easily generated, finally showing that the martensite content of the linking neck part and the ball center part is almost equivalent. The martensite distribution inside the steel ball is relatively uniform.

## 5. Conclusions

In this study, we investigated the microstructure evolution of steel balls during the cold skew rolling process via a multiscale constitutive model coupling martensite and grain refinement. The main conclusions drawn are as follows:Based on the mixing rule, a multiscale constitutive model of 316L SS was established by coupling martensitic transformation and grain refinement. By embedding the developed model into the FE software Simufact 16.0, a numerical simulation model of the cold skew rolling process of small-diameter 316L SS steel balls was developed.The simulation results were compared with the experimental results to verify the reliability of the established simulation model. It is found that the diameter size, dislocation density, grain size, and martensite content of the steel balls are in good agreement, which proves the prediction capability of the established model.The dislocation density in the large deformation of the steel balls is high. Consequently, the grain size in there is relatively small, which was determined through the analysis of the numerical simulation of microstructure evolution. Observing from the transverse section of the steel ball, the martensite content exhibited a slight difference between the areas in the surface and core. The martensite content in the longitudinal section of the steel ball was uniformly distributed in each region.

## Figures and Tables

**Figure 1 materials-16-03246-f001:**
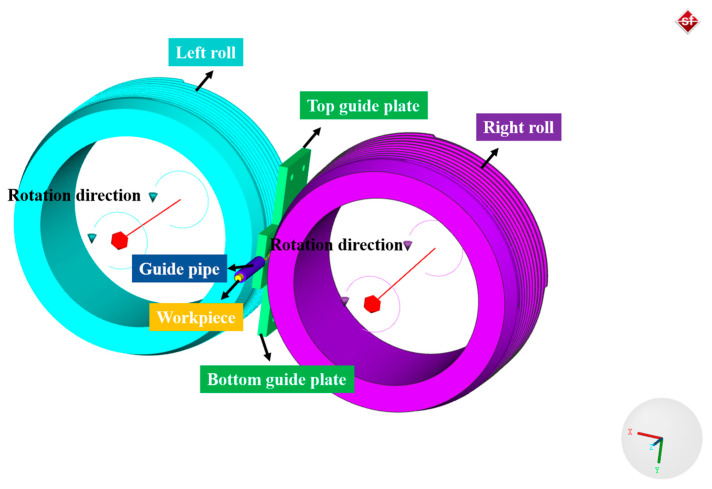
FE model of the cold skew rolling process of steel balls.

**Figure 2 materials-16-03246-f002:**
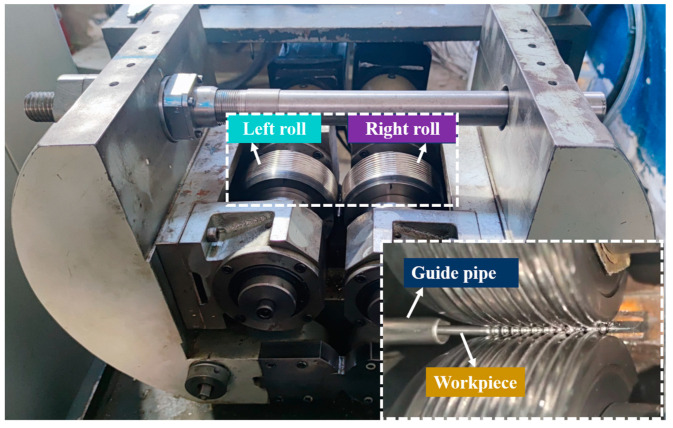
Experimental skew rolling mill and workpiece.

**Figure 3 materials-16-03246-f003:**
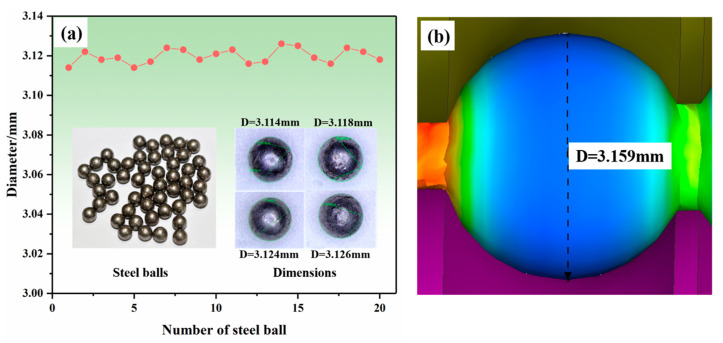
Comparison of the diameter values of steel balls: (**a**) experimental results and (**b**) simulation results.

**Figure 4 materials-16-03246-f004:**
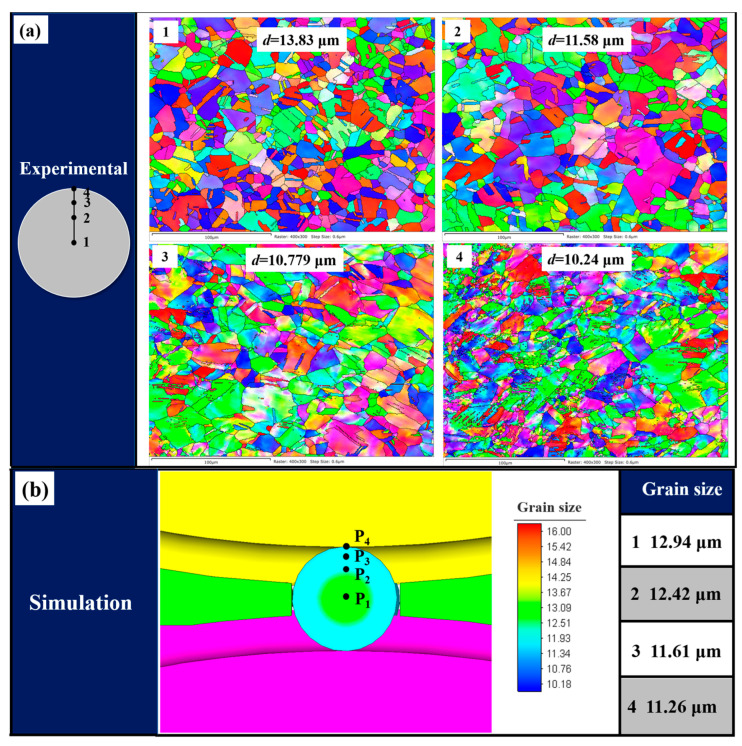
Comparison of grain size at different positions in transverse section of steel balls: (**a**) experimental results and (**b**) simulation results.

**Figure 5 materials-16-03246-f005:**
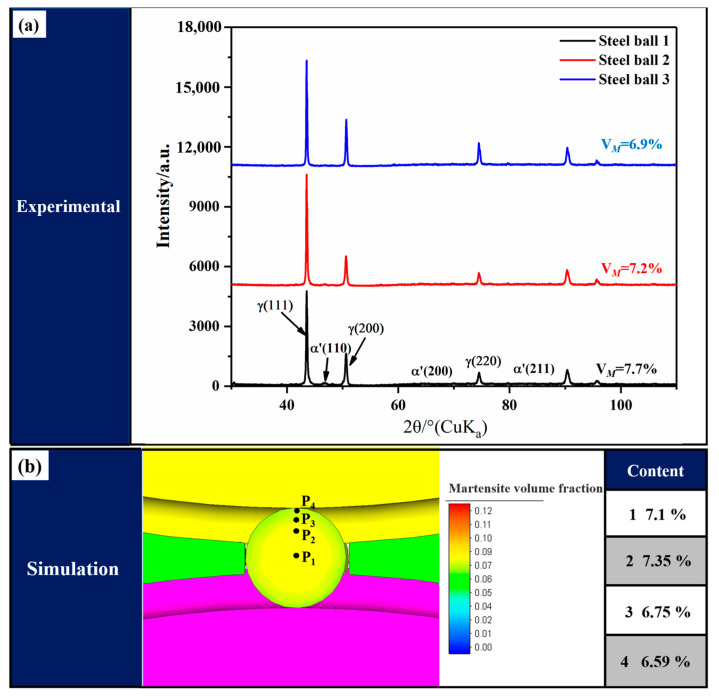
Comparison of volume fraction of martensite: (**a**) experimental results and (**b**) simulation results.

**Figure 6 materials-16-03246-f006:**
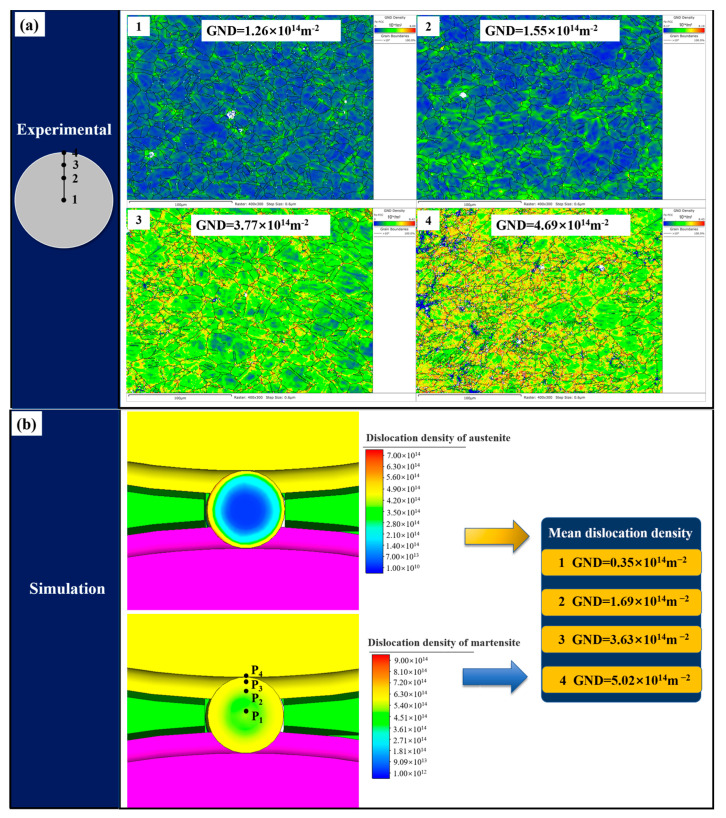
Comparison of dislocation density at different positions in transverse section of steel balls: (**a**) experimental results and (**b**) simulation results.

**Figure 7 materials-16-03246-f007:**
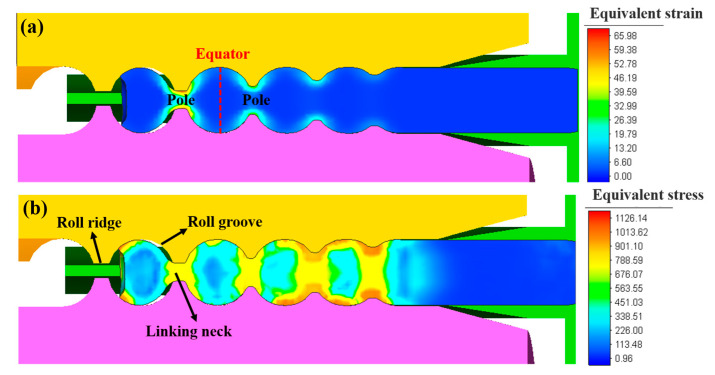
The strain and stress contour along the longitudinal section of the steel ball: (**a**) equivalent strain and (**b**) equivalent stress.

**Figure 8 materials-16-03246-f008:**
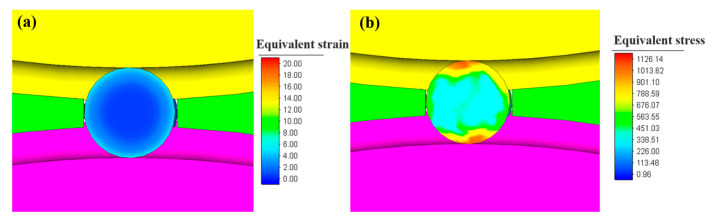
The strain and stress contour along the transverse section of the steel ball: (**a**) equivalent strain and (**b**) equivalent stress.

**Figure 9 materials-16-03246-f009:**
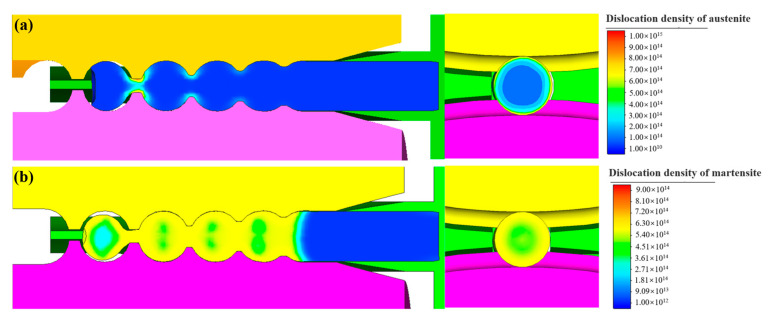
The dislocation density contour of steel balls (**a**) in austenite and (**b**) in martensite.

**Figure 10 materials-16-03246-f010:**
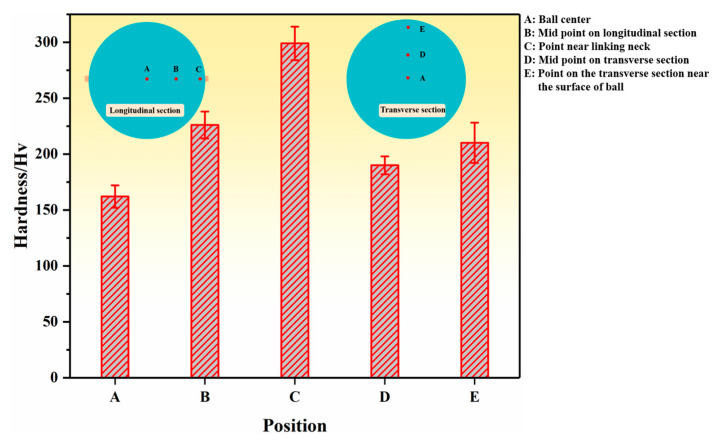
Microhardness of different positions in steel balls.

**Figure 11 materials-16-03246-f011:**
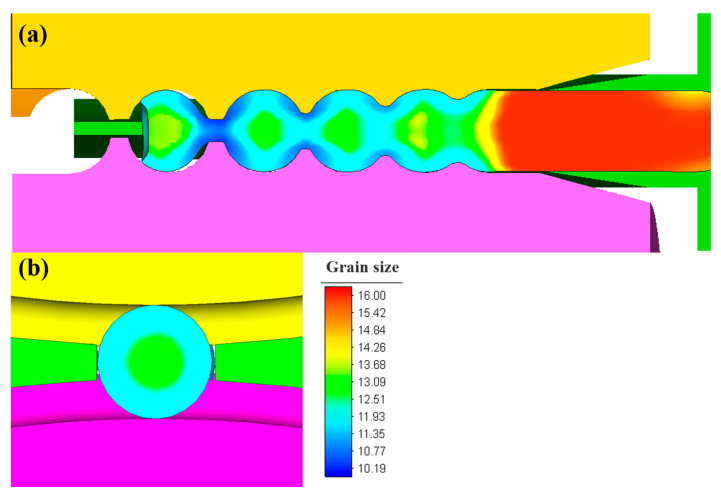
Grain size distribution of steel balls during the skew rolling process in (**a**) the longitudinal section and (**b**) the transverse section.

**Figure 12 materials-16-03246-f012:**
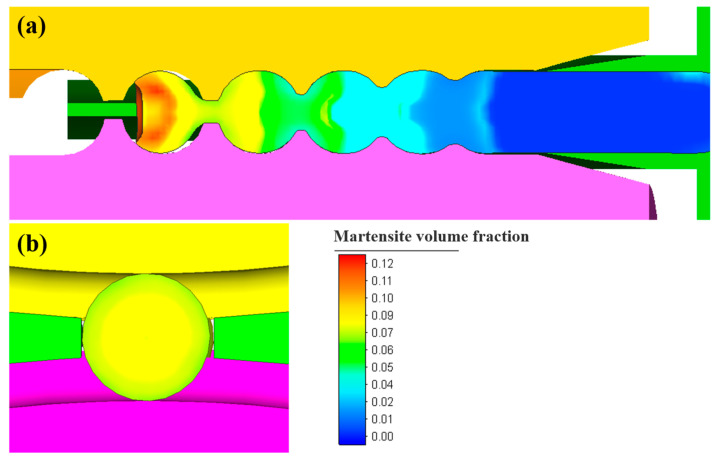
Martensite volume fraction of steel balls during skew rolling process in (**a**) the longitudinal section and (**b**) the transverse section.

**Table 1 materials-16-03246-t001:** Identified parameters of 316L SS constitutive model.

Parameters	Values	Parameters	Values
$\sigma_{y_{0}}$ (MPa)	1.0000 ×10^1^	$\sigma_{tr}^{0}$ (MPa)	7.0746 × 10^2^
$K_{s}$ (MPa·μm^1/2^)	6.5222 × 10^2^	$K_{tr}$ (MPa·μm^1/2^)	1.1347 × 10^2^
Ω	1.0515 × 10^−1^	$k_{a}$	4.1500 × 10^1^
$k_{d}$	2.1464 × 10^−6^	${\dot{N}}_{0}$	3.1500 × 10^17^
$k_{s}$	3.9676 × 10^1^	$d_{a}$	1.5088 × 10^−1^
$k_{M}$	1.5520 × 10^3^	$d_{b}$	4.1151 × 10^−1^
*m*	2.4255 × 10^2^	*K*	1.0000 × 10^5^
$c_{m}$	1.1090 × 10^−1^	α	4.0200 × 10^−1^
$c_{\gamma }$	7.7166 × 10^6^	$b_{M}$ (nm)	1.4700 × 10^−1^
$\sigma_{0}$ (MPa)	6.8790 × 10^2^	M	3.06 × 10^0^
$b_{A}$ (nm)	2.5600 × 10^−1^	$G_{A}$ (MPa)	7.5000 × 10^4^
$t_{M}$ (μm)	1.1000 × 10^0^	$G_{M}$ (MPa)	8.0000 × 10^4^

**Table 2 materials-16-03246-t002:** Chemical composition of 316L SS.

C	Mn	Si	Cr	Ni	Mo	Fe
0.03	1.74	0.27	16.82	10.26	2.08	Bal.

## Data Availability

The data used to support the findings of this study are available from the corresponding author upon request.

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
