# Peer review of "Numerical Prediction of Microstructure Evolution of Small-Diameter Stainless Steel Balls during Cold Skew Rolling"

_materials, 2023, doi:10.3390/ma16083246_

Round 1

Reviewer 1 Report

Page 7...line 180-181...grammatically incorrect

figure 4, how did you measure grain size in simulation??

figure 5, how the martensite volume fraction was evaluated in simulation?

Please clarify

figure 6, how the dislocation density is evaluated in simulation? its not clear...

Figure 9...same issue, how did it compute the dislocation density in FEA tool?  please add the description of the FEA tool and methodology, input parameters etc.

it this is based on FEA mesh, where is the mesh? its shown nowhere

Looks like the tool is throwing some results in some format, which is believed and shown here. What is the validation of the tool results?

What are the material models used in through FEA tool, what inputs are given? I think FEA methodology need to be elaborated.

Are you adjusting inputs to FEA tool to match the experimental results?

Such type of work seems to be already done as per the available literature, novelty is not explained properly.

Please elaborate on novelty of this work, how it is different that the others who have done it before

Author Response

Point 1: Page 7...line 180-181...grammatically incorrect

Response 1: The wrong sentence has been corrected. Please see Page 7 Line 180-181.

Point 2: figure 4, how did you measure grain size in simulation??

Response 2: The developed FE model is coupled with grain refinement, dislocation slip, and martensitic transformation. We carried out secondary development in the FE software Simufact and predicted the grain size, martensite content, and dislocation density variation within the skew rolling process. The grain size, martensite content, and dislocation density of the steel ball can be output in the post-processing interface. In addition, the grain size of different regions in the steel ball can be obtained by measuring the node’s value, as shown in Fig. 1.

Fig. 1 Grain size of steel ball center

Point 3: figure 5, how the martensite volume fraction was evaluated in simulation?

Response 3: During the simulation, each step’s strain increment and strain rate are known. According to the strain rate, the dislocation density change rate of martensite and austenite can be calculated, the critical stress of martensitic transformation occurring and the force acting on martensite can be obtained, and then the change rate of martensite volume fraction can be obtained. Finally, the martensite volume fraction of the steel ball can be obtained by iteration of the Euler algorithm.

Point 4: figure 6, how the dislocation density is evaluated in simulation? its not clear...

Response 4: Same as the previous question, as the strain rate is known, the evolution of austenite and martensite dislocation density can be calculated by the following equations. The dislocation density can be obtained by iteration of the Euler algorithm.

Point 5: Figure 9...same issue, how did it compute the dislocation density in FEA tool? please add the description of the FEA tool and methodology, input parameters etc. it this is based on FEA mesh, where is the mesh? its shown nowhere. Looks like the tool is throwing some results in some format, which is believed and shown here. What is the validation of the tool results? What are the material models used in through FEA tool, what inputs are given? I think FEA methodology need to be elaborated.

Response 5: As mentioned above, this paper uses the established multi-scale constitutive model to replace the material model in the FE software. The input parameters include the initial dislocation density, grain size, and martensite content of the steel ball, as well as the boundary conditions of the cold rolling process of the steel ball. We have added it to the revised manuscript, Page 5, line 165-167.

The grain size, dislocation density, and martensite content of the steel ball during the rolling process can be directly output in the post-processing interface. In order to observe the microstructure content distribution of steel balls more intuitively, this paper has hidden the mesh. The grain size distribution of steel ball with mesh is given in Fig. 2. The verification of the FEM results is provided in section 3.3. The simulation results of the macroscopic size and microstructure content of the steel ball are in good agreement with the experimental results, verifying the credibility of the established finite element model.

Fig. 2 Grain size of steel ball (with mesh)

Point 6: Are you adjusting inputs to FEA tool to match the experimental results?

Response 6: We did not modify the inputs to match the experimental results. The input variables are the initial grain size, dislocation density, and martensite content. The initial grain size of the steel ball is measured from the microstructure observation, as shown in Fig. 3. The initial dislocation density of the material is obtained from the reference [1]. Since the initial structure of SS316L is austenite, the initial martensite content is 0.

[1] Liu, S.; Li, W.; Shen, J.; Yang, X.; Wang, B.; Liu, J. Size-Dependent Constitutive Model Incorporating Grain Refinement and Martensitic Transformation. Arch. Civ. Mech. Eng. 2023, 23 (1).

Fig. 3 Initial grain size of 316L SS

Point 7: Such type of work seems to be already done as per the available literature, novelty is not explained properly. Please elaborate on novelty of this work, how it is different that the others who have done it before

Response 7: Previous research on steel ball rolling has focused primarily on the hot rolling of large-diameter steel balls, while there needs to be more research on the cold rolling of small-diameter steel balls. The existing cold rolling of small-diameter steel balls mainly analyzes the influence of process parameters on forming process and the analysis of stress, strain, and rolling force during the rolling process, lacking prediction of internal microstructure evolution. Therefore, this paper analyzes the microstructure evolution of small-diameter steel balls during the cold rolling process through the established FEM coupled with multiple deformation mechanisms.

Reviewer 2 Report

This paper present FE model that simulates the mechanical and microstructural behavior of  SS316 ball produced by a cold skew rolling process.  The FE model seems to be well developed to capture all the important events in the microstructure during compression/deformation, and the results are in good agreement with experimental data.  Also, the manuscript is very well written to provide an easy readability.   Yet, this paper is not without questions, and they are mostly related to the model.  

According to the experimental data (such as the in Fig.4), there is a significant level of grain refinement.  Since the rolling is done at room T, refinement by dynamic recrystallization is not likely.  Then, it should be resulted by nucleation of martensite phase.  Then, the question is how distribution of martensite phase seems more or less chaotic (how to predict where the new phase forms in simulation?).  Also, it seems that considerable level of twinning paralleled with martensitic transformation (Fig.4-1 and 2).  Plastic deformation by twinning may be energetically more favorable than by martensitic transformation.  The letter would be favored only in case when the strain is high.   The interplay between twining and martensite formation is not easy to capture correctly and it seems to be missing this paper, making the paper a bit confusing.

Author Response

Point 1: According to the experimental data (such as the in Fig.4), there is a significant level of grain refinement. Since the rolling is done at room T, refinement by dynamic recrystallization is not likely. Then, it should be resulted by nucleation of martensite phase. Then, the question is how distribution of martensite phase seems more or less chaotic (how to predict where the new phase forms in simulation?).

Response 1: The method used in this paper can only predict the content and overall distribution of martensite in different regions during the rolling process of steel balls. It cannot indicate the position of the newly formed martensite, as you thought. The position and morphology of newly formed martensite can be predicted by Phase field models or RVE and other methods. Still, the problem is that the calculation and time required are undoubtedly massive. In terms of solving engineering problems, the technique used in this paper is more convenient. As you mentioned, predicting the morphology and position of newly formed martensite during rolling is significant. We will conduct more in-depth research about this in the future.

Point 2: Also, it seems that considerable level of twinning paralleled with martensitic transformation (Fig.4-1 and 2). Plastic deformation by twinning may be energetically more favorable than by martensitic transformation. The letter would be favored only in case when the strain is high. The interplay between twining and martensite formation is not easy to capture correctly and it seems to be missing this paper, making the paper a bit confusing.

Response 2: We are very grateful for your suggestion. The initial structure of 316L stainless steel contains a portion of annealing twins. The annealed twin is generally larger in shape and has parallel boundaries, which usually run through the grain. Since the location of Fig.4-1 and 2 is in the center of the ball, the amount of deformation is relatively small, and the twins here have the morphological characteristics of annealed twins, so the twins here are annealed twins. Admittedly, deformation-induced twins do occur in stainless steel during plastic deformation, but the content of deformation twins produced is smaller compared to martensite, so the effect of deformation twins on the work-hardening of the material can be neglected, as that mentioned in references [1] and [2].

[1] Meng, B.; Liu, Y.Z.; Wan, M.; Fu, M.W. A Multiscale Constitutive Model Coupled with Martensitic Transformation Kinetics for Micro-Scaled Plastic Deformation of Metastable Metal Foils. Int. J. Mech. Sci. 2021, 106503.

[2] Hamasaki, H.; Ohno, T.; Nakano, T.; Ishimaru, E. Modelling of Cyclic Plasticity and Martensitic Transformation for Type 304 Austenitic Stainless Steel. Int. J. Mech. Sci. 2018, 146-147, 536-543

Round 2

Reviewer 1 Report

Ok, but can still be improved

Reviewer 2 Report

Replies are acceptable.